# Long-Term Care Services and Insurance System in China: An Evolutionary Game Analysis

**Han Hu** [1,*,†] **and Zhao Zhang** [2,*,†]

1   School of Public Policy and Administration, Xi'an Jiaotong University, Xi'an 710028, China
2   School of Economics and Finance, Xi'an Jiaotong University, Xi'an 710061, China
*   Correspondence: huhan112@xjtu.edu.cn (H.H.); zhzhxjtu@xjtu.edu.cn (Z.Z.)
†   These authors contributed equally to this work.

**Abstract:** The study creates a simplified game model to propose a suitable policy to foster a win-win scenario between care institutions and families of the disabled elderly, and to give a reference basis for enhancing the welfare level of the disabled elderly as well as the commercial performance of care institutions. First, we study and synthesise the experiences of contemporary Chinese long-term care pilot cities to offer data for subsequent numerical analysis; second, we create an evolutionary game model to depict the conflicts and evolving patterns of conflicts between the disabled elderly and care facilities in China; and third, we use numerical analysis to investigate the effects of internal factors (cost of care, price of care) and policy assistance (government subsidies). Finally, we utilise numerical analysis to investigate how internal factors (cost of care, price of care) and policy assistance (government subsidy) affect the combination of solutions. The study reveals that (1) the two-sided strategic choices of care institutions and disabled families make the game unstable. (2) The government can influence the choice of care institutions and disabled families by tax rates and subsidies, implying a stability policy. (3) The presence of an evolutionary stabilization strategy implies that government control may have a desirable limit. When government engagement in this market is limited, "professional care, home care" finally becomes the evolutionary stability method.

**Keywords:** long-term care; long-term care insurance; care facilities; disabled families; evolutionary game

## 1. Introduction

Population ageing and accelerated ageing are the hallmarks of China's demographic change [1,2]. In China, the prevalence of disorders such myocardial infarction, malignant tumours, stroke-related complications, hypertension, diabetes mellitus, dementia, and deafness has been rising annually in recent years. Consequently, the individual or family suffers a tremendous burden of care and financial strain in the event of long-term care hazards [3], and, for severely incapacitated individuals, continuity. In China, the long-term care insurance policy plan and regulating mechanisms are incomplete, and a range of long-term care models are in the pilot stage. Given their diseases or limitations, the majority of elderly individuals prefer to maintain their independence and dignity, remaining in their own homes and participating in their communities to obtain care services, if possible. Over the past generation, however, there has been a significant shift: children are less likely to care for an elderly parent, and elderly parents are more likely to live alone for longer and to reside in an institution such as a nursing home. Thus, the practise of long-term care insurance models in China's pilot cities can be loosely categorised as "institution care model" and "in-home care model". Research and surveys have shown that there is a conflict between care institutions and families with disabilities; care institutions expect to set prices for care services that are well above cost, while families with disabled, elderly members prefer low prices and high-quality services. In order to investigate the behavioural mechanisms underlying their mutual collaboration, and to identify win-win tactics for the

successful implementation of long-term care insurance policies, this research focuses on the cooperative behaviour of care institutions and disabled families in long-term care.

## 2. Literature Review

### 2.1. Long-Term Care Insurance

The financing practice of long-term care insurance can be traced back to the 1970s, according to whether the local long-term care insurance system has been established. It can be divided into "supplement mode" and "universal benefit mode". The "supplement mode" refers to the mode of providing care services and subsidies based on household surveys in regions where long-term care insurance system has not been established [4]. The "universal benefit mode" refers to the mode of providing services and subsidies based on the system covering the whole population in regions where insurance is independently established. "Inclusive mode" can be divided into "tax mode", represented by Nordic countries and mainly funded by fiscal revenue, and "insurance mode", funded by social insurance payment [5,6].

In general, many nations and regions have opted for a policy-based long-term care insurance model, while the United States represents nations with a commercial insurance model. As a developing country that ages before becoming wealthy, China can only establish a social insurance-based long-term care insurance system. The lessons learned from relevant countries and regions, whether in the design of the system framework (e.g., the funding method, the level of care, and how to pay) or the design of the system's components, provide a wealth of information for our country. However, there are also significant variations in the content and implementation of the models across countries. The comparison reveals significant differences between the systems in Germany, Japan, and the United States, including vast disparities in system coverage, different levels of individual contribution burden, varying levels of system detail, and different approaches to cost control and professional caregiver training (see Table 1).

**Table 1.** Comparison of long-term care insurance in Germany, Japan, and the United States.

| | Germany | Japan | United States |
|---|---|---|---|
| System Coverage | Covers nearly all nationalities | Only covers half the country's citizens | Individuals' volitional participation |
| Individual financial burden | Government shoulders one-third of the burden, while businesses and individuals shoulder the remaining fifty percent | Individuals bear 10% first, followed by 50% of the remaining 90% | The age, duration of benefits, waiting period, and scope of liability vary among insurance companies |
| Hierarchy of system design | Grade III | Grade VI | —— |
| Cost control | Introduction of competitive mechanisms for service providers and fund managers; implementation of uncompensated caregivers | Increasing the use of volunteers to provide low-cost care services; instituting preventive care to limit expected cost increases | Insurance companies to maximise benefits, spontaneously adopt cost-cutting methods, and focus on insurance actuarial |
| Nursing staff training | The mechanism for cultivation consists primarily of secondary education and the establishment of a variety of specialties | Depending on the developed domestic volunteer system, the supply is inadequate and the staffing gap is significant | Insurance companies have numerous options based on the training system of medical personnel |

Source: Based on the relevant literature.

### 2.2. The Game of Long-Term Care

There exists an unbalanced game of interests between the designated care stations, the government, and disabled families. In the information asymmetry or silos, care institutions are likely to take advantage of their absolute resources to deceive commissioners, specif-

ically by deceiving the government and families to obtain policy and financial support through providing low-skilled services or cutting corners in the implementation of the LTC insurance policy [7]. In addition, the care stations' revenues are related to the care services, so they may induce excessive medical services and demand to increase the revenues. Overconsumption of medical resources will increase the utility of more comprehensive and nuanced services for an insured individual, who can easily develop a propensity for overconsumption. All of these behaviours can lead to misuse of health care resources, market distortions, and a reduction in social welfare. According to Snorre Kverndokk (2021), some countries have implemented a fee to prevent bed blockage in hospitals [8]. This paper introduces the Stackelberg game model, which positions hospitals as leaders and care providers as followers. It then examines the impact of such a fee on the strategic decisions of hospitals and long-term care providers, and theorises that the average level of hospital services will be close to its lowest level prior to the reform.

The responsibility for providing home care for the elderly, also known as elder care, falls on the family in traditional civilizations, because it is the smallest social unit. However, as a result of economic development and changes in family structure, the shrinkage of the family has become a significant social feature in China, and the traditional family's function of caring for the handicapped elderly is decreasing, particularly the professionalism and complexity of the care demands of the severely disabled elderly [9–11]. This results in institutional care becoming a trend. The Ministry of Health Insurance issued the "Guidance on Expanding the Pilot Long-term Care Insurance System" in 2020, stating unequivocally that priority should be given to stimulating the development of home care services as well as making full use of social care facilities to provide long-term care services for the disabled elderly at home (as shown in Figure 1).

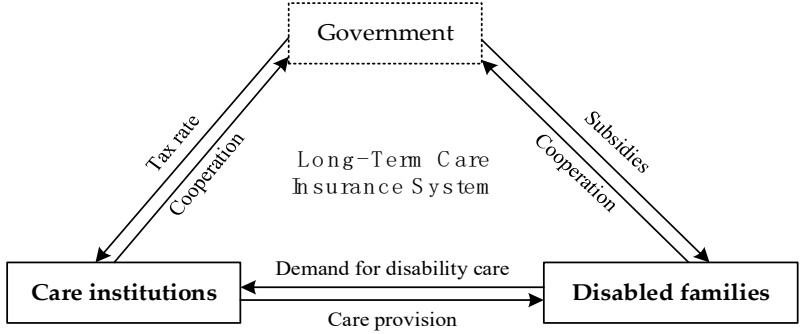

**Figure 1.** Relationship between the government, care facilities, and disabled families (Chinese context).

It is estimated that the disabled elderly population in China will exceed 100 million in 2030 and reach a peak of 129 million around 2050. Compared with the short-term acute medical expenses of the disabled elderly, the long-term care expenses increased rapidly [12]. In 2016, the Ministry of Human Resources and Social Security issued the "Guidance on the Piloting of Long-term Care Insurance System", which launched the piloting of long-term care insurance in 15 cities, including Qingdao, Guangzhou, and Chengdu. In September 2020, the state expanded the scope of the piloting again, and 14 cities, including Tianjin and Kunming, entered the second batch of piloting. By the end of June 2019, the National Medical Insurance Administration had counted 88.54 million participants in fifteen pilot cities and two key provinces, with 426,000 people receiving benefits and over RMB 9200 in annual fund payments. According to the Statistical Bulletin on the Development of National Medical Security in 2020, there were 4845 designated care institutions for long-term care insurance, with 191,000 nursing staff.

The pilot cities have developed different policies based on local realities in terms of care model and payment level, which have produced different effects. The practice of long care insurance models in China's pilot cities can be roughly divided into the "in-institution care model", which is further divided into designated care facilities and hospitals, and the

"in-home care model", which is further divided into full day care and part-time care [13,14], as shown in Table 2.

**Table 2.** Security groups, payment criteria, and limits for long-term care insurance in 21 pilot cities.

| City/Province | Covered Groups | Long Term Care Models | |
|---|---|---|---|
| Chengde/Hebei | Severe disability | In-institution care<br>In-home care<br>Subsidy for home care services | |
| Changchun/Jilin | Severely disabled, terminal cancer palliative care patients | In-institution care (long-term stay)<br>In-institution care (short-term stay) | |
| Qiqihar/Heilongjiang | Severe disability | In-home care (full day)<br>In-home care (part-time)<br>In-institution care | |
| Shanghai | 60 years old and above with a level 2–6 disability | In-institution care (hospital)<br>In-institution care<br>In-home care | |
| Nantong/Jiangsu | Severe disability, moderate dementia | In-institution care (hospital)<br>In-institution care<br>In-home care | |
| Suzhou/Jiangsu | Severe and moderate disability | In-institution care<br>In-home care | |
| Ningbo/Zhejiang | Severe disability | professional care<br>In-hospital care | |
| Anqing/Anhui | Severe disability | In-institution care (hospital)<br>In-institution care<br>In-home care | |
| Shangrao/Jiangxi | Severe disability | Self-care<br>Door-to-door care<br>Product rental<br>In-institution care | |
| Qingdao/Shandong | Totally disabled, severe dementia patients | Disability | Specialist care<br>Home care<br>Patrol<br>Long-term care |
| | | Dementia | Short-term care<br>Day care |
| Jinmen/Hubei | Severe disability | In-home care (full day)<br>In-home care (part-time)<br>In-institution care<br>In-institution care (hospital) | |
| Guangzhou/Guangdong | Long-term disability, extended care, equipment users | In-institution care<br>In-home care<br>In-institution care (hospital) | |
| Chongqing | Severe disability | In-institution care<br>In-home care services | |
| Shihezi City/Xinjiang | Severe disability | Agreed institutions/home care<br>Non-Agreed | |
| Shijingshan District/Beijing | Severe disability | In-institution care<br>Door-to-door care<br>In-home care | |
| Tianjin | Severe disability | In-institution care<br>In-home care | |

**Table 2.** *Cont.*

| City/Province | Covered Groups | Long Term Care Models |
|---|---|---|
| Kaifeng/Henan | Severe disability | In-institution care<br>In-home care<br>Self-care |
| Nanning/Guangxi Zhuang Autonomous Region | Severe disability | In-institution care<br>Off-site services<br>In-home care (full day)<br>In-home care (part-time) |
| Kunming/Yunnan | Severely disabled and dementia patients | In-institution care (hospital)<br>In-institution care<br>In-home care |
| Hanzhong/Shaanxi | Severe disability | In-institution care (hospital)<br>In-institution care<br>In-home care (full-day)<br>In-home care (part-day) |
| Urumqi/Xinjiang Uyghur Autonomous Region | Severe disability | In-home care (full-day)<br>In-home care (part-day)<br>In-institutional care (full-day) |

## 3. Model and Methods

Before constructing an evolutionary game model in order to analyse the game and strategy choices of nursing institutions and families with disabilities, we first make the following assumptions based on China's realistic background and economic common sense.

**A1**: Assumes that the care institution provides two kinds of care services: simple care services and professional care services.

Nursing institutions, as market participants, meet the general characteristics of enterprises. In China's long-term care market, a variety of nursing services can be divided into simple care services and professional care services. When providing simple nursing services, nursing institutions can obtain lower business revenue with lower investment costs. Nursing institutions can obtain higher business revenue when providing professional services, but this also means hiring more nursing staff and purchasing more nursing equipment. From the perspective of economics, nursing institutions will choose the most favourable strategy based on cost-benefit analysis [15].

**A2**: Assumes that families with disabilities can choose in-home services or in-institution services, both of which supplied by care institution.

The choice of nursing services depends not only on the degree of disability of the disabled elderly, but also on the time cost of care, income status, and satisfaction with nursing services of the families of patients with disabilities. Choosing in-home services can also reduce the family's long-term care expenditure, for which family members have to bear a higher cost of care time—which may also become an opportunity cost that restricts the family income level. Choosing in-institution services will reduce the time cost of care for family members; however, high-quality care services can also be obtained by paying for higher long-term care expenditure [16,17]. Families need to consider the above factors comprehensively in order to choose the most suitable strategy for themselves.

**A3**: Although the government does not directly participate in the game between the families and the care institutions, it can provide subsidies for the care institutions by taxing all the families.

The operation of long-term care institutions requires a certain amount of nursing staff and nursing equipment in the early stage, which means that there is a certain threshold for the operation of such institutions. In addition, only when the business revenue from nursing services (or the number of disabled elderly people receiving care) reaches a certain level can nursing institutions balance their income and expenditure, and thus gain profits. Without

the policy support of government departments, the operation of nursing institutions may be in trouble [18–20]. Therefore, in the model analysis of this paper, we assume that the government departments can set different subsidy standards according to the operation mode of nursing institutions.

*3.1. Construction of Evolutionary Game Model*

1. Game subjects and strategies: The game subjects include the disabled family (i.e., a family containing a disabled older person) and the care institution. The game strategy for the family = {in-home care, in-institution care}. "In-home care" means the families take care of the disabled person at home and only purchase a portion of the services of the care institutions, which implies that the family may spend part of their time caring for the disabled person. "In-institution care" means that the incapacitated person lives in a care institution and the family does not have to devote time to care. The game strategy for care institutions = {simple care, professional care}. Simple care only provides basic care for daily living; 'professional care' relieves pain, alleviates illness, and prolongs life [21–23].

2. The role of government departments: the game is regulated through long-term care insurance and subsidies to care institutions.

3. Features of the game: The game constructed in this paper has three characteristics. First of all, it is a "one-to-many" asymmetric game, i.e., a care institution must play with multiple disabled households; secondly, the total number of households, *N*, and the proportion of disabled households (between 0 and 1) can be regarded as known parameters; thirdly, the calculation of insurance and subsidies involves summation.

4. Game gains and losses: the payoff matrix of the long-term care game is shown in Table 3.

**Table 3.** Payoff matrix for the long-term care game.

| Care Institution | Disabled Household | In-Home Services $(1-q)$ | In-Institution Services $(1-q)$ |
|---|---|---|---|
| Simple care services $(1-p)$ | | $((1-t)P_1 + \rho T_1 - C,$ $Y(1-\tau_1) - tX - (1-t)P_1)$ | $(P_1 + T_1' - C,$ $Y(1-\tau_1) - P_1 + W)$ |
| Professional care services $(p)$ | | $((1-t)P_2 + \rho T_2 - C(1+\delta),$ $Y(1-\tau_2) - tX - (1-t)P_2)$ | $(P_2 + T_2 - C(1+\delta),$ $Y(1-\tau_2) - P_2 + W)$ |

Note: It is assumed that all symbols in the above table represent positive numbers >0.

The main parameters in the game are described as follows.

$P_1$, the price of "simple care" offered by care institutions;

$P_2$, the price of "professional care" provided by care institutions (in general, assume $P_2 > P_1$);

$T_1$, government subsidies for care institutions under the "simple care" model;

$T_2$, government subsidies for care institutions under the "professional care" model (in general, assume $T_2 > T_1$);

$\rho$, the proportion of institutions receiving government subsidies under home care model (between 0 and 1) when compared to institutional care (this assumption is based on the limited services that provided by care institutions under the home care model, and the government also undertakes part of the public service functions of long-term care);

$C$, the cost when care institutions provide "simple care";

$\delta$, cost-plus rates for care institutions providing "professional care";

$Y$, pre-tax income of households;

$t$, the proportion of time spent caring for a disabled person under the home care model (the value ranges from 0 to 1, with the proportion of time spent by a disabled family purchasing care services in this model being $1 - t$);

$X$, cost per unit of care time when families choose in-home care;

$W$, increased benefits for disabled people when families choose in-institution care as compared to in-home care;

$\tau_1$, tax rates for long-term care insurance under the "simple care" model;

$\tau_2$, tax rates for long-term care insurance under the "professional care" model (in general, assume $\tau_2 > \tau_1$);

The government is not directly involved in the game, but it does play a moderating role. Long-term care insurance is mainly used for the care of disabled patients, so the above game implies a budgetary equilibrium: regardless of the type of model, the expenditures of long-term care insurance should be equal to the government subsidies to care institutions (i.e., the third feature of the game). Thus, the following equation should hold:

$$N\theta T_i = N \tau_i Y \tag{1}$$

In the above equation, $i = 1$ or 2, represents "simple care" or "professional care", respectively.

### 3.2. Replication Dynamics

The long-term care game model reflected in Table 2 implicitly suggests that the strategic choices of care institutions and disabled families are not stable. An evolutionary game analysis approach is adopted below to answer the above questions. It is assumed that care institutions adopt the "professional care" and "simple care" strategies with probabilities $p$ and $1 - p$, respectively, and that families with disabilities adopt the "in-institution care" and "in-home care" strategies, with probabilities $q$ and $1 - q$, respectively. Generally, it is assumed that $p, q \in [0, 1]$. The following is an analysis of the pay-offs for different strategies adopted by care institutions and disabled families in the game [22–25].

According to Table 2, the expected benefits of the professional care strategy for care institutions are:

$$u_1 = q[P_2 + T_2 - C(1 + \delta)] + (1 - q)[(1 - t)P_2 + \rho T_2 - C(1 + \delta)] \tag{2}$$

The expected benefits of simple care strategy for care institutions are:

$$u_2 = q(P_1 + T_1 - C) + (1 - q)((1 - t)P_1 + \rho T_1 - C) \tag{3}$$

Therefore, the expected average benefit for care institutions adopting these two different strategies is:

$$u = pu_1 + (1 - p)u_2 \tag{4}$$

Taylor and Jonker [4] have proposed the use of replicative dynamic equations to analyse strategic choices and their evolution in games [26]. According to Friedman (1991, 1998), the essence of replicating dynamic equations is a dynamic differential equation of the adoption frequency of a particular strategy [5,6]. The growth rate of the number of professional care strategies adopted by care institutions can be reflected by $u_1 - u$.

The growth rate of the numbers of professional care strategies adopted by care institutions can be expressed, so the replication dynamic equation for the choice of professional care strategy by care institutions is

$$\frac{dp}{dt} = F(p) = p(u_1 - u)$$
$$= p(1 - p)\{q[t(P_2 - P_1) + (1 - \rho)(T_2 - T_1)] + (1 - t)(P_2 - P_1) + \rho(T_2 - T_1) - \delta C\} \tag{5}$$

Substituting Equation (1) into the above equation, we can obtain:

$$\frac{dp}{dt} = F(p) = p(1 - p)(\alpha_1 q + \beta_1) \tag{6}$$

where $\alpha_1 = t(P_2 - P_1) + (1 - \rho)Y(\tau_2 - \tau_1)/\theta$, $\beta_1 = (1 - t)(P_2 - P_1) + \rho Y(\tau_2 - \tau_1)/\theta - \delta C$.

Equally, according to Table 2, the benefits of choosing the in-institution care strategy for disabled families are:

$$v_1 = p[Y(1 - \tau_2) - P_2 + W] + (1 - p)[Y(1 - \tau_1) - P_1 + W] \tag{7}$$

The expected benefits of adopting the in-home care strategy for disabled families are:

$$v_2 = p \cdot [Y(1 - \tau_2) - tX - (1 - t)P_2] + (1 - p)[Y(1 - \tau_1) - tX - (1 - t)P_1] \quad (8)$$

Therefore, the expected average return for a disabled family choosing these two different strategies is:

$$v = qv_1 + (1 - q)v_2 \quad (9)$$

Similarly, the growth rate of the number of institutional care strategies adopted by disabled families can also be expressed as $v_1 - v$. Thus, the replication dynamic equation for the choice of the 'in-institution care' strategy for disabled families is:

$$\frac{dq}{dt} = G(q) = q(v_1 - v) = q(1 - q)(-\alpha_2 p + \beta_2)$$
$$\alpha_2 = t(P_2 - P_1), \ \beta_2 = W + tX - tP_1 \quad (10)$$

In summary, the set of equations consisting of the replication dynamics, comprising Equations (6) and (10), reflects the replication dynamics system of the game between care institutions and disabled households, which reveals the dynamic trends in the probability of choosing different strategies between the two parties of the game. From Equations (6) and (10), the strategy choice between the two parties in the game is related to parameters such as the proportion of disabled families $\theta$, family income $Y$, family care time $t$, family care cost $X$, long-term care insurance tax rates ($\tau_1$ and $\tau_2$), and the cost of care services ($C$ and $\delta$). It is important to note that the time of family care $t$, and the price of long-term care services ($P_1$ and $P_2$) are key factors influencing the payoffs for both sides of the game, while the policy instruments of long-term care insurance ($\tau_1$, $\tau_2$ and $\rho$) can only unilaterally influence the strategic choices of care institutions.

*3.3. Analysis of Evolutionarily Stable Strategy*

Evolutionarily Stable Strategy (ESS) is adopted to further analyse the game reflected by the replicator dynamic system. For care institutions, the $F(p)$ in the replication dynamic equation reflected in (6) can be derived from $p$:

$$F'(p) = (1 - 2p)(\alpha_1 q + \beta_1) \quad (11)$$

The conditions for care institutions to choose an evolutionary stabilization strategy are [27]: $F(p) = 0$ and $F\prime(p) < 0$. Let $F(p) = 0$; we can solve for $p = 0$ or 1, and $\alpha_1 q + \beta_1 = 0$, that is:

$$q^* = [\delta C - (1 - t)(P_2 - P_1) - \rho Y(\tau_2 - \tau_1)/\theta]/[t(P_2 - P_1) + (1 - \rho)Y(\tau_2 - \tau_1)/\theta] \quad (12)$$

According to the conditions for the existence of the evolutionary stabilization strategy, when the probability $q = q^*$ for the families with disabilities to adopt the institutional care strategy, it is the evolutionary stabilization strategy of the care facilities to choose "professional care" with any probability $p$. When $q > q^*$, only $p = 1$ satisfies $F(p) = 0$ and $F\prime(p) < 0$; thus, $p = 1$ is the evolutionary stabilization strategy of care facilities. When $q < q^*$, only $p = 0$ satisfies $F(p) = 0$ and $F\prime(p) < 0$; thus, $p = 0$ is the evolutionary stabilization strategy of care facilities. Furthermore, when $q^*$ takes values between 0 and 1, there exists the following inequality:

$$(1 - t)(P_2 - P_1) + \frac{\rho Y(\tau_2 - \tau_1)}{\theta} < \delta C < P_2 - P_1 + \frac{Y(\tau_2 - \tau_1)}{\theta} \quad (13)$$

Similarly, for the disabled family, the derivative of $G(q)$ to $q$ in the dynamic equation reflected in Equation (10) can be obtained:

$$G'(p) = -(1 - 2q)(-\alpha_2 q + \beta_2) \quad (14)$$

Similarly, the conditions for the disabled family to choose evolutionary stability strategies are: $G(p) = 0$ and $G\prime(p) < 0$. Let $G(p) = 0$, then $q = 0$ or $1$, and $-\alpha_2 q + \beta_2 = 0$, namely:

$$p^* = [W + tX - tP_1]/[t(P_2 - P_1)] \tag{15}$$

According to the conditions for the existence of the evolutionary stabilization strategy, when the probability $p = p^*$ for care facilities to adopt the "professional care" strategy, the disabled family chooses "institutional care" with any probability. $q$ is the evolutionary stabilization strategy; when $p > p^*$, only $q = 0$ satisfies $G(p) = 0$ and $G\prime(p) < 0$; thus, $q = 0$ is the evolutionary stabilization strategy of the disabled family. In addition, when the value of $q^*$ ranges from 0 to 1, there exists the following inequality:

$$tP_1 < W + tX < tP_2 \tag{16}$$

To summarize, according to the replication dynamic system, if and only if $0 < p^* < 1$ and $0 < q^* < 1$ are true, the regulatory game model constructed in this paper has five Nash equilibrium points, namely $(0,0)$, $(0,1)$, $(1,0)$, $(1,1)$, and $(p^*,q^*)$, among which $p^*$ and $q^*$ are given by Equations (10) and (12).

According to Friedman (1991) [5], local stability analysis of the Jacobian matrix of a replicating dynamic system can be performed to identify the stability of the five Nash equilibrium points mentioned above. According to the definition of the Jacobian matrix, the Jacobian matrix of the replicated dynamic system reflected by Equations (4) and (8) is:

$$J = \begin{vmatrix} \frac{\partial F(p)}{\partial p} & \frac{\partial F(p)}{\partial q} \\ \frac{\partial G(q)}{\partial p} & \frac{\partial G(q)}{\partial q} \end{vmatrix} = \begin{vmatrix} (1-2p)(\alpha_1 q + \beta_1) & p(1-p)\alpha_1 \\ -q(1-q)\alpha_2 & (1-2q)(-\alpha_2 p + \beta_2) \end{vmatrix} \tag{17}$$

The equilibrium point is substituted into the Jacobian matrix. If the determinant $det(J) > 0$ and trace $tr(J) < 0$ are satisfied, the equilibrium point is the local asymptotic stable fixed point of the evolutionary game, namely, the evolutionary stable strategy. The stability analysis of the above five Nash equilibrium points is based on the local stability analysis of the Jacobi matrix, and the results are shown in Table 4.

**Table 4.** Stability analysis of equilibrium point.

| Equilibrium Point | Det (J) | Tr (J) | Conditions | Result |
|---|---|---|---|---|
| (0, 0) | + | − | $\beta_1 < 0, \ \beta_2 < 0$ | ESS strategy |
| | indefinite | indefinite | —— | Saddle point |
| (0, 1) | + | − | $\alpha_1 + \beta_1 < 0, \ \beta_2 > 0$ | ESS strategy |
| | indefinite | indefinite | —— | Saddle point |
| (1, 0) | + | − | $\beta_1 > 0, \ -\alpha_2 + \beta_2 < 0$ | ESS strategy |
| | indefinite | indefinite | —— | Saddle point |
| (1, 1) | + | − | $\alpha_1 + \beta_1 > 0, \ -\alpha_2 + \beta_2 > 0$ | ESS strategy |
| | indefinite | indefinite | other | Saddle point |
| $(p^*, q^*)$ | indefinite | 0 | —— | central point |

(1) Simple care services, in-home services: The condition of this evolutionary stability strategy is $\beta_1 < 0$, $\beta_2 < 0$. $\beta_1 < 0$ means $(1-t)(P_2 - P_1) + \rho Y(\tau_2 - \tau_1)/\theta < \delta C$, which reflects that the extra benefit of "professional care" compared with "simple care" is lower than the corresponding extra cost, which causes care institutions to give up the "professional care" strategy. $\beta_2 < 0$ means $W < t(P_1 - X)$, which reflects that the increased benefits for the disabled household by choosing "institutional care" are less than the savings by

choosing "home care", so the disabled household tends to choose "home care" in order to reduce the actual expenditure (i.e., $(1 - t)P_1$).

(2) Simple care services, in-institution services: The condition of this evolutionary stability strategy is $\alpha_1 + \beta_1 < 0$, $\beta_2 > 0$. $\alpha_1 + \beta_1 < 0$ means $P_2 - P_1 + \frac{Y(\tau_2 - \tau_1)}{\theta} < \delta C$, which reflects that the extra cost of "professional care" is high compared with "simple care", which causes care institutions to give up the "professional care" strategy. $\beta_1 > 0$ means $W > t(P_1 - X)$, which reflects that the increased welfare of disabled families choosing "institutional care" is higher than the cost saved by choosing "home care", so disabled families tend to choose "institutional care".

(3) Professional care services, in-home services: The condition of this evolutionary stability strategy is $\beta_1 > 0$, $-\alpha_2 + \beta_2 < 0$. $\alpha_1 + \beta_1$ means $(1 - t)(P_2 - P_1) + \rho Y(\tau_2 - \tau_1)/\theta > \delta C$, which reflects that the extra benefit of "professional nursing" is higher than the corresponding extra cost of "simple nursing", which causes care institutions to tend to choose the "professional nursing" strategy. $-\alpha_2 + \beta_2 < 0$ means $W < t(P_2 - X)$, which reflects that the increase in benefits of the disabled family choosing "institutional care" is lesser than the cost savings of choosing "home care", so the disabled family tends to choose "home care" to reduce the actual expenditure (i.e., $(1 - t)P_2$).

(4) Professional care services, in-institution services: The condition of this evolutionary stability strategy is that $\alpha_1 + \beta_1 > 0$, $-\alpha_2 + \beta_2 > 0$. $\alpha_1 + \beta_1 > 0$ means $P_2 - P_1 + \frac{Y(\tau_2 - \tau_1)}{\theta} > \delta C$, which reflects that the extra cost of "professional nursing" is low enough compared with "simple nursing", which causes care institutions to tend to choose the "professional nursing" strategy. $-\alpha_2 + \beta_2 > 0$ means $W > t(P_2 - X)$, which reflects that the disabled family's selection of "in-institution care" caused their welfare to increase to a higher degree than their savings would by choosing "home care", so the disabled family tends to choose the "institutional care" strategy first.

## 4. Numerical Simulation Analysis

### 4.1. Parameter Setting

Assuming N is normalized to 1, at present, the disabled elderly population in China exceeds 40 million (See http://www.nhc.gov.cn/xcs/s3574/202112/141e8f26e8c24c93b89e27202da41b51.shtml (accessed on 9 December 2021)), which accounts for about 0.02857 of China's total population (1.4 billion). Therefore, we assume that $\theta = 0.02857$. Since China does not currently have a unified long-term care insurance system, we can assume that the value of the parameters $\tau_1$ and $\tau_2$ will not exceed the current medical insurance payment ratio. We assume $\tau_1 = 0.02$, $\tau_2 = 0.04$, and in addition, we assume $\rho = 0.5$.

We conducted a survey of typical Chinese cities that have experimented with long-term care insurance in order to define suitable parameters for the evolutionary game model. The Chinese Older Care Services survey (CECS), which was conducted by the School of Public Administration at Xi'an Jiaotong University in 2021, is the primary source of data on elderly persons with disabilities and long-term care insurance. The sample was selected as shown in Figure 2.

Based on this survey, we obtain $P_1$ = RMB 2847.19 per month for simple care and $P_2$ = RMB 3289.14 per month for professional care. For the care institution, the simple cost of caring for the disabled elderly is RMB 3200 per month. Considering the additional cost of professional care, we set $\delta$ to 0.8. Through the survey, we found that the average annual income of disabled families is RMB 64,755.37, which means $Y$ = RMB 5396.28 per month. In the survey, we found that the average length of time that the disabled family cares for the elderly under the condition of home care is 0.4406. Under home care, the disabled family saves some additional expenses (such as the wages of hiring caregivers), and the corresponding care costs are lower than institutional care. Therefore, $X$ is set to RMB 2000. However, professional nursing can improve the health status of the disabled elderly, which not only improves their living conditions, but also does not crowd out the working time of disabled families. Therefore, $W$ is set to RMB 500. The settings of these parameters are reported in Table 5.

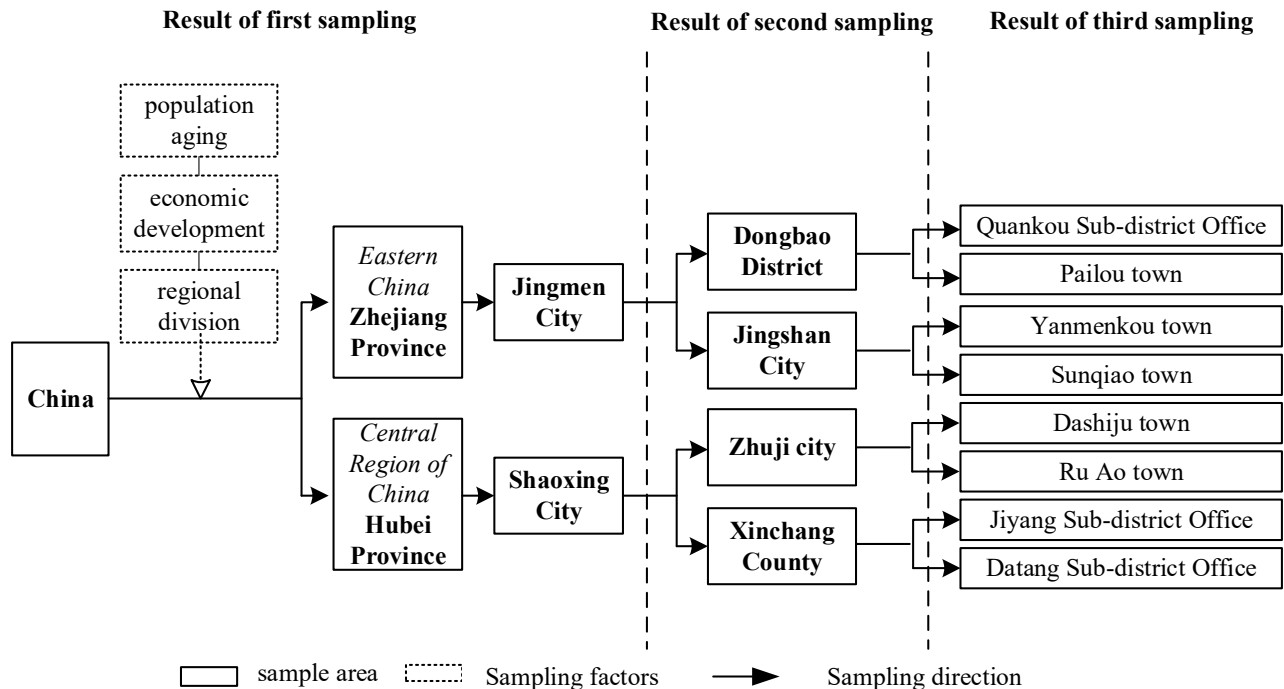

**Figure 2.** Sample selection.

**Table 5.** The settings of the game parameters.

| Participants in the Game | Key Variables | Variable Symbols | Variable Value |
|---|---|---|---|
| Care facility | Price of simple care | $P_1$ | 2847.19 |
| | Price of professional care | $P_2$ | 3289.14 |
| | Cost of simple care | $C$ | 3200 |
| | Cost-plus rate of professional care | $\delta$ | 0.80 |
| Care institution | Household income before taxes | $Y$ | 5396.28 |
| | Time ratio of family care for disabled people | $t$ | 0.4406 |
| | Unit time cost of family care for disabled people | $X$ | 2000 |
| | Increased family benefits in institutional care | $W$ | 500 |

The parameter settings in Table 4 are relatively representative of the actual situation in China. Based on these settings, we can calculate the profit matrix of the game participants under the benchmark conditions, as shown in Table 5. It can be seen from Table 5 that there is no pure-strategy Nash equilibrium in this closing matrix. Therefore, in the subsequent analysis, we focused on the conditions under which ESS appeared by adjusting some parameter values (see Table 6).

**Table 6.** Payoff matrix of long-term care game.

| Care Institution / Disabled Household | In-Home Services $(1 - q)$ | In-Institution Services $(q)$ |
|---|---|---|
| Simple care services $(1 - p)$ | (281.51, 2814.44) | (3424.77, 2941.16) |
| Professional care services $(p)$ | (−142.47, 2459.38) | (5084.31, 2391.29) |

*4.2. Benchmarking*

According to the settings in Table 2, combined with Equations (12) and (15), we can find $p^* = 0.651$, $q^* = 0.203$ (Figure 3). When at least one of the initial probabilities ($p$)

of care institutions choosing "professional care" and the initial probability ($q$) of disabled families choosing "institutional care" is not at the centre point, the strategy selection probabilities of both sides fluctuate around the centre point, as shown in Figure 1. Only when the initial probabilities of $p$ and $q$ are located at their respective centre points does the probability of strategy selection not change with time. In the subsequent analysis, we assume that the initial probabilities of $p$ and $q$ are 0.5.

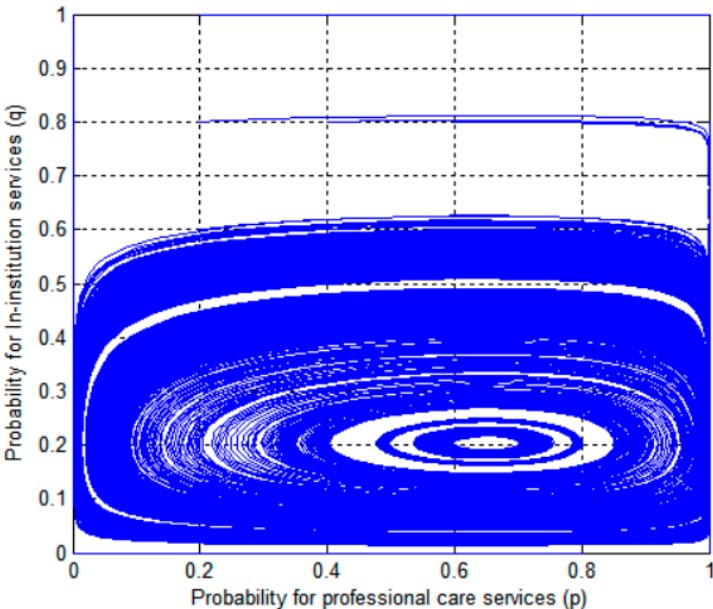

**Figure 3.** The phase diagram of the game between care providers and families with disabilities.

Considering that the initial willingness $p$ of care institutions to adopt the "professional care" strategy and the initial willingness $q$ of disabled families to adopt the "institutional care" strategy will have a great impact on the game outcome, there are four scenarios selected here to analyse the impact of the cooperative game outcome between the initial willingness of care institutions and disabled families, as follows (see Figure 4).

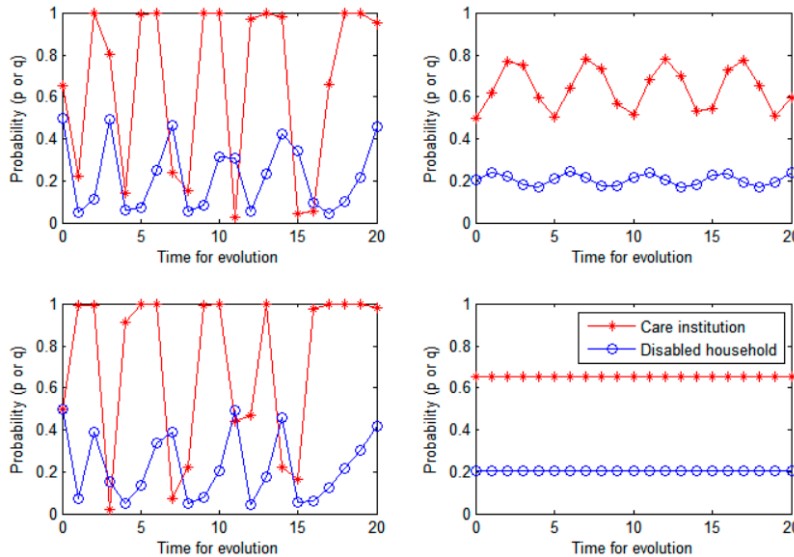

**Figure 4.** The effect of the initial willingness of care facilities and disabled families on the evolutionary path of the game.

(1) The initial willingness to adopt the "professional care" strategy was high in care institutions, as was the initial willingness to adopt the "institutional care" strategy in families with disabilities, i.e., $p = p^*$ and $q = 0.5$.

(2) The initial willingness of the care institution to adopt the "professional care" strategy was low, and the initial willingness of the disabled families to adopt the "institutional care" strategy was also low, i.e., $p = 0.5$ and $q = q^*$.

(3) The initial willingness to adopt the "professional care" strategy in care institutions was low, whereas the initial willingness of families with disabilities to adopt the "institutional care" strategy was high, i.e., $p = 0.5$, $q = 0.5$.

(4) The initial willingness to adopt the "professional care" strategy was high in care institutions, whereas the initial willingness of families to adopt the "institutional care" strategy was low, i.e., $p = p^*$ and $q = q^*$.

## 5. Results

### 5.1. Costs of Simple Care and Cost-Plus Rate of Care

For care institutions, when the cost *C* of "simple care" was low, care institutions tended to choose the "professional care" strategy. When the cost *C* of "simple care" was high, care facilities tended to choose the "simple care" strategy [28–30]. For disabled families, when the cost *C* of "simple care" was low, care facilities tended to choose the "home care" strategy; when the cost *C* of "simple care" was high, care facilities tended to choose the "institutional care" strategy (see Figure 5).

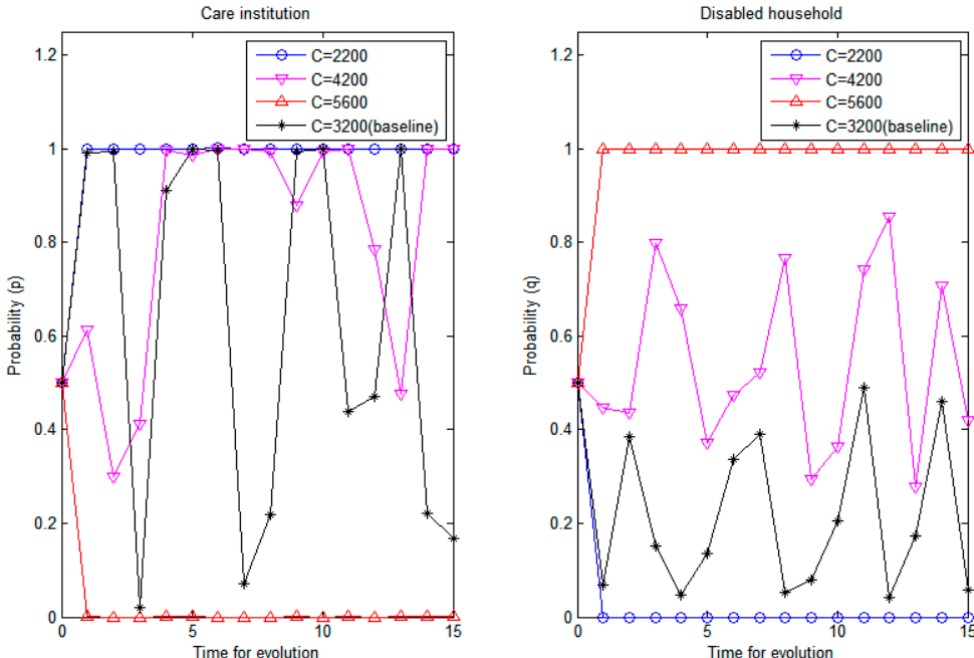

**Figure 5.** The effect of costs of simple care on the evolutionary path of probabilities *p* and *q*.

For care facilities, when the cost-plus $\delta$ for professional care was low, care facilities tended to choose the professional care strategy; when the cost-plus $\delta$ for "simple care" was high, care facilities tended to choose the "simple care" strategy. For disabled families, when the cost-plus $\delta$ for simple care was low, care facilities preferred the home care strategy; when the cost-plus $\delta$ for simple care was high, care facilities tended to choose the "institutional care" strategy (see Figure 6).

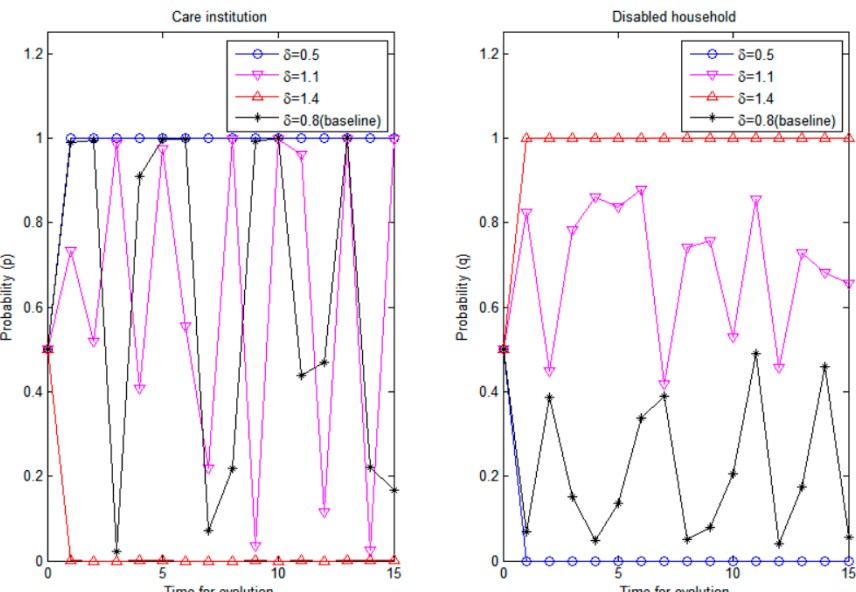

**Figure 6.** The effect of cost-plus of care on the evolutionary path of probabilities *p* and *q*.

### 5.2. Cost and Time Required of Home Care

Families with disabilities preferred the "in-home care" strategy when the cost of home care was low in order to reduce the cost of care, and the "in-institutional care" strategy when the cost of home care was high in order to reduce the cost of care. For care institutions, they preferred the "simple care" strategy when the cost of home care was low, and the "professional care" strategy when the cost of home care was high. In practice, "simple care, in-home care" is more suitable for families with less severe disability, and is more like an initial stage in the development of long-term care. In this combination, the cost of care at home is relatively low, so the care services demand is not very high, and "in-home care" can already meet the patient's needs [31–33]. For care institutions, there is also a preference for the "simple care" strategy, as the demand is low, and the provision of "professional care" may be priced out of the market (see Figure 7).

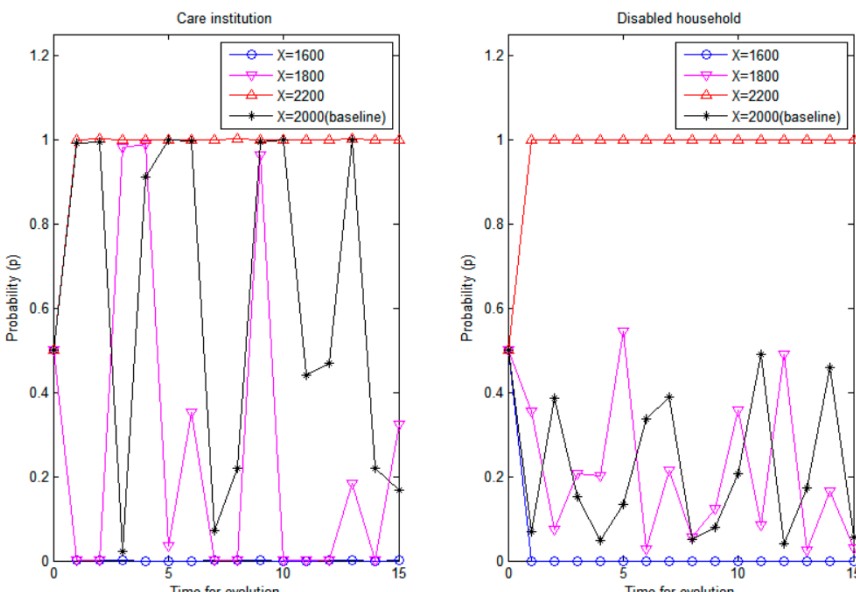

**Figure 7.** The effect of cost of home care on the evolutionary path of probabilities *p* and *q*.

Disabled families preferred the "in-institution care" strategy when the time required for home care was low, and the "in-home care" strategy when the time required for home care was high. For care facilities, they preferred the professional care strategy when the time required for home care was low and the simple care strategy when the time required for home care was high. When the time required for home care is low, it means that the savings from "in-home care" are lower, while the choice of "in-institution care" increases the welfare of the disabled elderly by providing them with better care conditions, i.e., $W > t(P_i - X)(i = 1 \text{ or } 2)$, which leads the family to choose "in-institution care". In this case, care institutions tend to choose the strategy of "professional care" because of its greater benefits. When the time required for home care is higher, this means that the savings from "home care" are significant enough to compensate for the lost welfare from giving up "in-institution care", i.e., $W < t(P_i - X)(i = 1 \text{ or } 2)$, which leads to disabled families choosing "home care". In this case, care institutions choose "professional care" with extra benefits lower than the corresponding extra costs, so they tend to give up this strategy and choose the "simple care" strategy (see Figure 8).

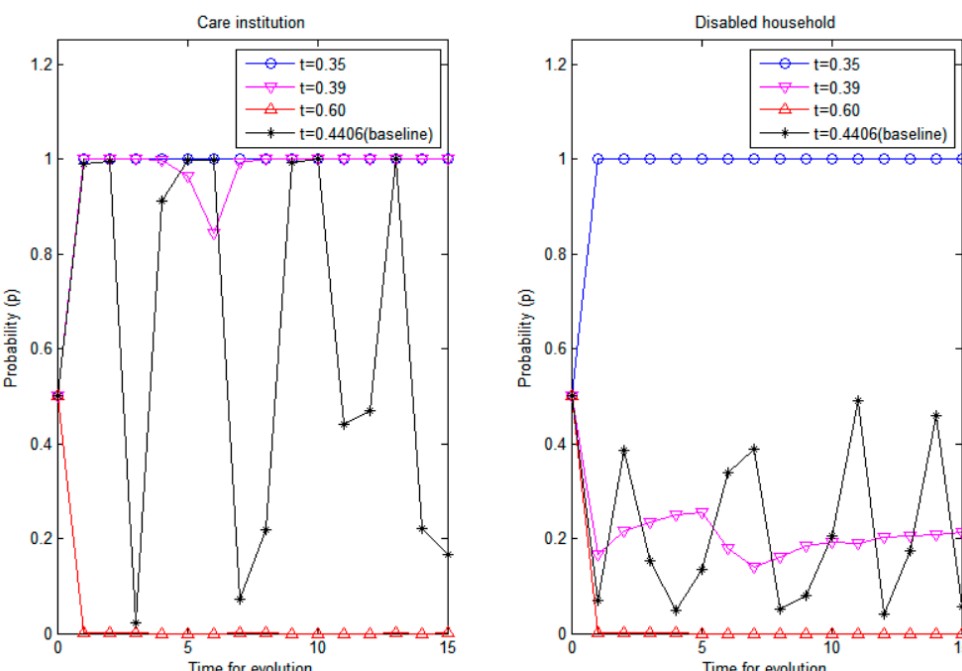

**Figure 8.** The effect of time required for home care on the evolutionary path of probabilities *p* and *q*.

### 5.3. Prices of Simple Care and Professional Care

For care institutions, when the price of "simple care" was low, they tended to choose "professional care"; when the price of "simple care" was higher, they tended to choose "simple care". For disabled families, there was a preference for "home care" whether "simple care" is underpriced or overpriced. Care institutions may lose money when the price of "simple care" is underpriced, so only when the price of "simple care" is higher than "professional care" will this strategy become the dominant strategy of care institutions. The strategy selection of disabled families is largely influenced by the strategy selection of care institutions; when the price of "simple care" is low, the disabled family tends to realize that care facilities will adopt the strategy of "professional care", so choosing "home care" will incur lower nursing expenses [34,35]. When the price of "simple care" is high, the disabled family tends to realize that the care facilities will adopt this strategy, and will still choose "home care" to reduce the cost of care (see Figure 9).

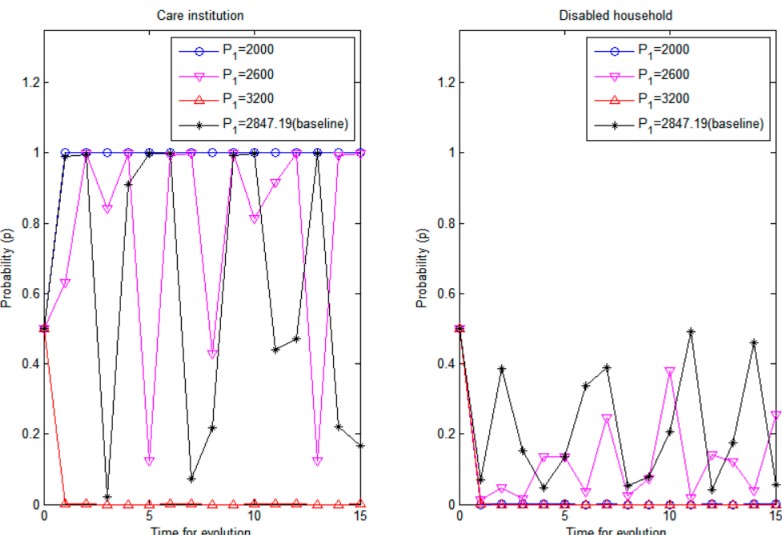

**Figure 9.** The effect of the price of simple care on the evolutionary path of probabilities *p* and *q*.

For disabled families, there was a preference for "institutional care" when the price of professional care was too low, and a preference for "home care" when the price of "professional care" was too high. For care facilities, the preferred option was professional care whether the price of professional care is underpriced or overpriced. Nursing facilities may also face losses when "professional care" is underpriced. However, since the price of "professional care" was still higher than the price of "simple care", nursing institutions tended to choose the "professional care" strategy to reduce losses. For disabled families, the low price of "professional care" tended to be within their affordable range, and as this strategy could also increase the welfare of the disabled elderly, it became a dominant strategy. When the price of "professional care" was too high, it was able to bring more benefits to nursing institutions, so it became a dominant strategy. However, in this scenario, the disabled families could no longer afford the expensive nursing services, so they were more inclined to choose "home care" to reduce the actual price (i.e., $(1-t)P_2$) (see Figure 10).

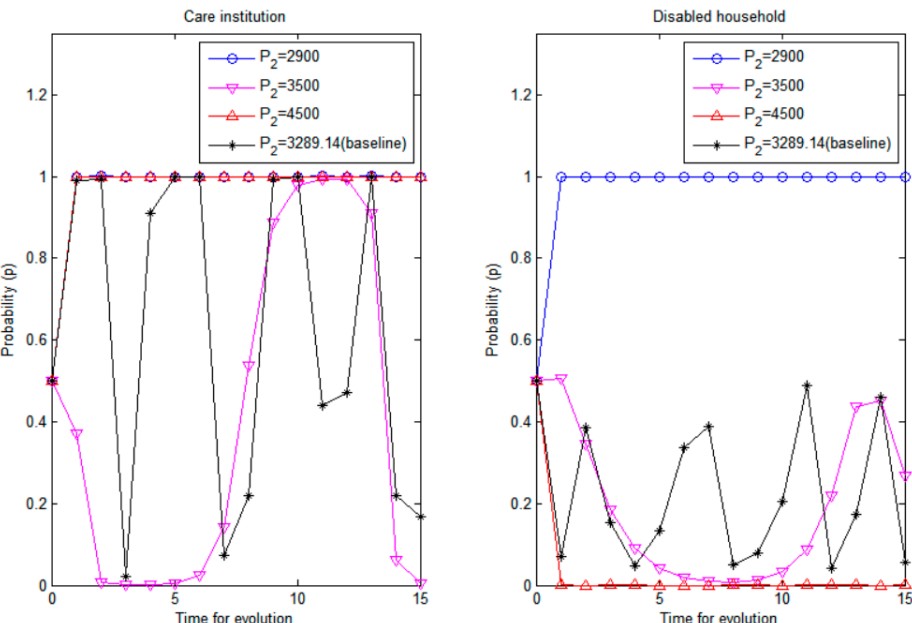

**Figure 10.** The effect of price of professional care on the evolutionary path of probabilities *p* and *q*.

## 6. Further Discussion: Long-Term Care Insurance System

The government, as a major player in the development of long-term care home care services, actively participates in the integration of resources in the construction of long-term home care service systems on the one hand, but on the other hand, it also needs to monitor and regulate the healthy development of long-term care service systems by formulating necessary tax policies and subsidy policies.

### 6.1. Tax for Simple Care Services

The lower the tax rate under the "simple care" model, the more likely care institutions are to choose "skilled care" and the more likely families are to choose "home care"; the higher the tax rate under the "simple care" model, the more likely care institutions are to choose "simple care" and the more likely families are to choose "institutional care".

When the tax rate is lower under the "simple care" model, under balanced budget conditions, it means that the long-term care insurance subsidy for the simple care model is not sufficient to cover the costs. Thus, care facilities tend to choose the highly subsidized "professional care". In this scenario, disabled families tend to give up "institutional care" to save the burden of care. The higher the tax rate under the "simple care" model, the higher the government subsidy under balanced budget conditions, which will induce care facilities to adopt this strategy. At the same time, due to the lower price of simple care, the family will likely select "institutional care" in order to obtain additional benefits for the disabled elderly within their budget (see Figure 11).

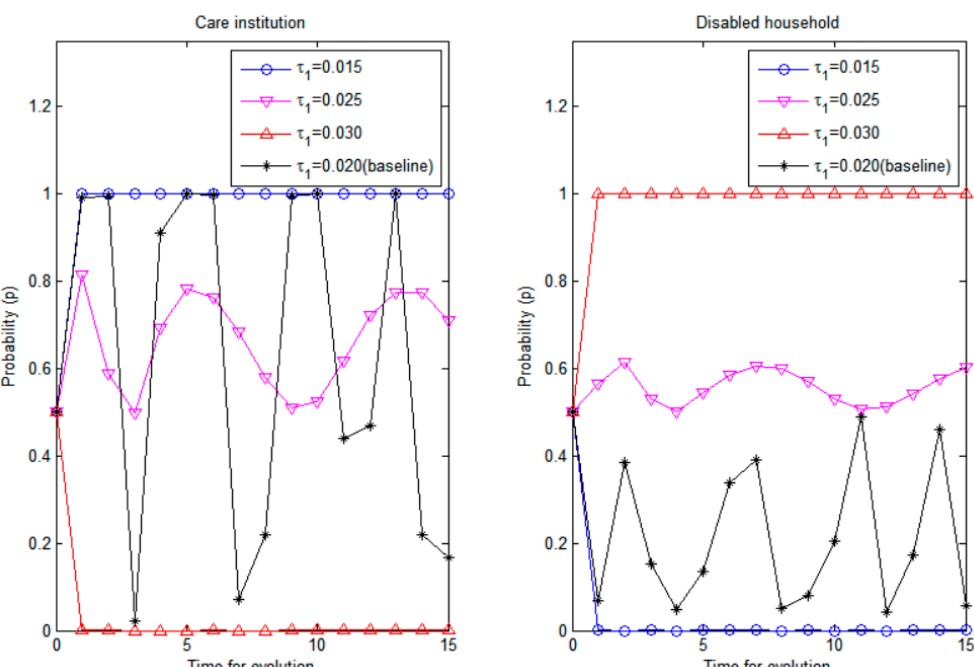

**Figure 11.** The tax for simple care on the evolutionary path of probabilities *p* and *q*.

### 6.2. Tax for Professional Care Services

The lower the tax rate under the professional care model, the more care facilities are willing to choose "simple care" and the more disabled families are willing to choose "institutional care"; the higher the tax rate under the professional care model, the more care facilities tend to choose "professional care" and the more disabled families tend to choose "home care" (see Figure 12).

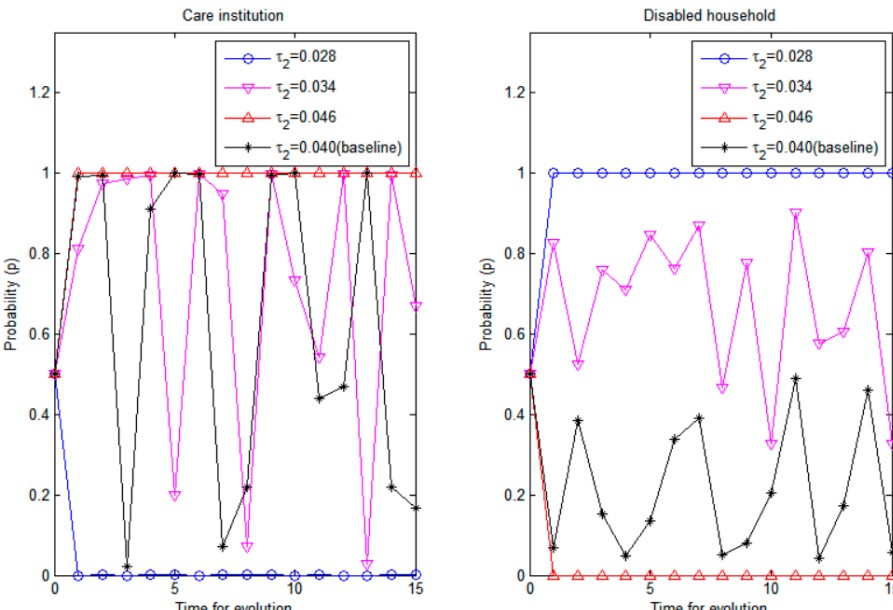

**Figure 12.** The effect of tax for professional care on the evolutionary path of probabilities *p* and *q*.

The lower the tax rate under the professional care model, the lower the subsidy from the long-term care insurance system for the professional care model in a balanced budget condition, which means that the benefits of the model may not be sufficient to cover the costs, so care facilities tend to abandon this strategy in favour of the simple care strategy. At this point, families will choose the "institutional care" strategy in order to obtain additional benefits for the disabled elderly, given that the price of simple care is lower and within their reach. "The higher the tax rate under the professional care model, the higher the government subsidy under balanced budget conditions, which may induce care facilities to choose this strategy; at this time, families may prefer to drop institutional care in order to save the burden of care".

### 6.3. Subsidies Proportion for In-Home Care Services

Since "home care" is a simpler model than "institutional care", and costs less in manpower and resources, care facilities only send people to the home to provide limited "simple care" or "professional care" services. Consequently, the government subsidies to care facilities are discounted compared to institutional care when families choose the "home care" strategy (noted as $\rho$ in this paper). Obviously, the value of $\rho$ affects the strategic choice of both sides of the game (see Figure 13).

The higher the government subsidy for the home care model, the more likely it seems that disabled families will choose "home care" and care facilities will choose professional care. When the government subsidies for the home care model are low, it indicates that the government does not support this model. In this case, the trend of the game will be affected by a variety of factors; thus, it will be unstable. On the contrary, the higher the government subsidies for the home care model, the more the government appears to support this model. When the value of $\rho$ is close to 1, it indicates that the disabled families tend to choose "home care" or "institutional care", and that the subsidies received by nursing institutions are similar, meaning that the government's intervention in this market is weak. At this point, the direction of the game in the long-term care insurance market is more dependent on the parties involved. Thus, for care facilities, the choice of professional care is more profitable, and for disabled families, the choice of home care saves them the burden of care [36]. Ultimately, these models (professional care, home care) become an evolutionary stabilization strategy.

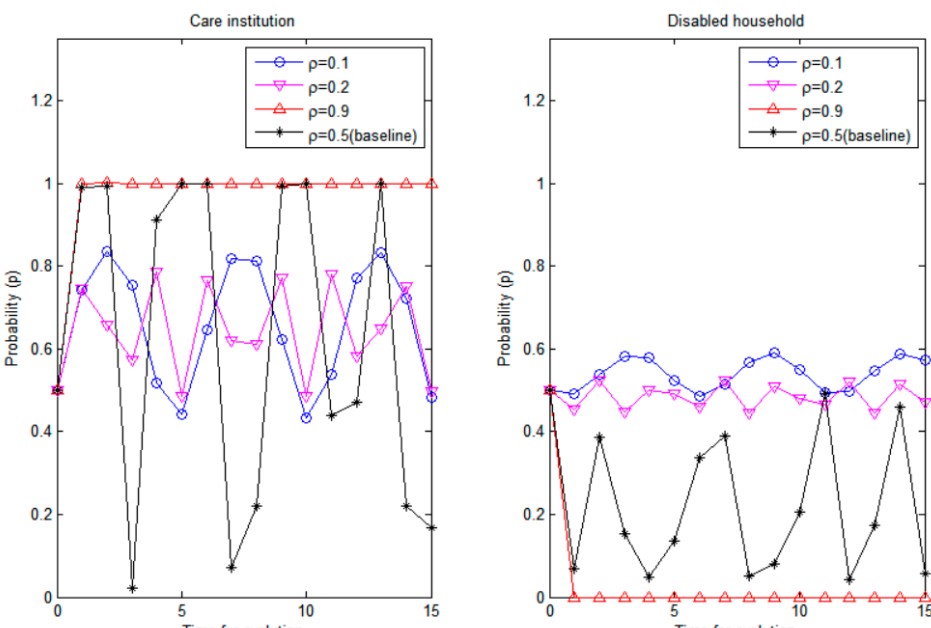

**Figure 13.** The effect of subsidies proportion for in-home care on the evolutionary path of probabilities $p$ and $q$.

## 7. Conclusions and Suggestions

### 7.1. Conclusions

First, the two-sided nature of the strategic choices of care institutions and disabled families leaves the game unstable. On the one hand, care institutions may select simple care or professional care to maximize their profitability, whereas on the other hand, families with disabilities may choose between home and institutional care to maximize their welfare. In some cases, there is no evolutionary stabilization strategy in the game between care institutions and disabled families, and the strategic choices of both sides fluctuate around the central point [37].

Second, the government can influence the choice of care institutions and disabled families through tax rates and subsidies, which implies the existence of an evolutionary stabilization strategy. The lower the tax rate of "simple care", the more often care providers choose professional care as the preferred strategy. The more disabled families choose "home care" as the preferred strategy, the lower the tax rate of "professional care", and the more often care facilities choose "simple care". The lower the tax rate for "professional care", the more often care providers choose "simple care" as the preferred strategy, and the more often disabled families choose "institutional care" as the preferred strategy. The closer the government subsidy for home care is to institutional care, the more often care providers choose "professional care" as the preferred strategy, and the more often disabled families choose "institutional care" as the preferred strategy. The closer the government subsidies for home care are to those for institutional care, the more often care providers will choose "professional care" as the dominant strategy. In this scenario, "home care" will become the dominant strategy for families with disabilities.

Third, the existence of an evolutionary stabilization strategy predicts that government regulation may have a desirable boundary. When government intervention in this market is weak, "professional care, home care" eventually becomes the evolutionary stabilization strategy.

### 7.2. Suggestions

The following particular measures are proposed based on the realities of care institutions and home care services in China's LTC insurance pilot areas to standardise the quality of care services in the pilot phase, to protect the rights and interests of persons with

disabilities, and to promote a good interaction between LTC insurance-designated care stations and families with disabilities.

First, LTC insurance payments must be made more flexible in order to save money. Many pilot regions currently do not differentiate between handicap levels and employ a single payment reimbursement rate, which does not promote the logical use of LTC insurance resources (e.g., Table 1). For example, the level of need for long-term care for Level 2 and Level 6 handicapped seniors is not the same, and with the same payment reimbursement rate, it is simple for Level 2 disabled seniors to conspire with nursing stations and utilise LTC insurance resources to provide services other than nursing care, resulting in the phenomena of low-quality nursing care driving out high-quality care.

Second, government regulation and oversight should be improved to purify the long-term care operating environment. Care facilities should be fined and/or suspended from receiving government funding if they fail to deliver the necessary care services. Families of elderly people with disabilities should also face consequences for switching care facilities in an unreliable manner.

Third, the knowledge asymmetry among the primary actors in long-term home care services should be reduced. For instance, we should incorporate long-term care insurance services into the elderly community services section so that residents can learn about long-term care in an accessible and understandable way. We should ask the designated long-term care insurance care stations in the area to conduct lectures in the community so that the elderly may comprehend the benefits of each care station's services and maximise the degree of matching between the care stations and families with disabilities.

Fourth, caregiver professionalism should be increased. Care stations that can only provide basic care have fewer professional caregivers for three reasons: first, the overall number of professional caregivers is small; second, the size of the care stations is tiny, making it difficult to recruit professional caregivers; and third, there is no retraining mechanism for caregivers. The number of professional carers can be increased via numerous measures, such as encouraging universities to offer long-term care majors, adding long-term care training in job training, and providing a retraining mechanism for caregivers in care facilities.

**Author Contributions:** H.H. and Z.Z. designed the study, performed the statistical analysis, and drafted & revised the manuscript. All authors have read and agreed to the published version of the manuscript.

**Funding:** This study was funded by National Social Science Foundation of China (No 22CRK014), Key Grant Project of Chinese Ministry of Education (No. 18JZD045), Chinese Postdoctoral Science Foundation (No.2019M663773) and The National Social Science Fund of China (No.22CRK014).

**Institutional Review Board Statement:** The study was conducted in accordance with the Declaration of Helsinki, and approved by the Institutional Review Board (or Ethics Committee) of Xi'an Jiaotong University (protocol code: 2018-1200; date of approval: 16 November 2018) for studies involving humans.

**Informed Consent Statement:** Not applicable for this survey only collects the cost of home care and institutional care for the disabled elderly, and does not involve other human studies.

**Data Availability Statement:** The data that support the findings of this study are available from the authors H.H. & Z.Z., upon reasonable request.

**Conflicts of Interest:** The authors declare no conflict of interest.

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
