# Peer review of "Long-Term Care Services and Insurance System in China: An Evolutionary Game Analysis"

_sustainability, doi:10.3390/su15010610_

Round 1

Reviewer 1 Report

Giving congratulations to the researchers has captivated me the rigor and dedication to the work presented. Just make a suggestion to update the citations of more than 5 years present in the text. for this I recommend the review of the most current literature.

Author Response

Your suggestions helped us revise and improve our work, and they served as a valuable guide for our study. We've updated the literature review and bibliographic index, and we'd like to thank you again for your help.

Reviewer 2 Report

Remarks:

1) In the introduction, the main goal and specific goals, research problems and research hypotheses were not clearly defined.

2) The bibliography is quite modest and does not include health care in other countries.

3) What do A1, A2, A3 mean? In the chapter MODEL AND METHODS? Why did the authors fail to describe and develop these assumptions?

4) No review of the analyzed statistical data

5) It is difficult to say how the research was done to make the data reliable? Are the data samples representative? "Data on elderly people with disabilities and long-term care insurance are primarily drawn from the Chinese Elderly Care Services survey (CECS), which was conducted by School of Public Administration of Xi'an Jiaotong University in 2021. The sample was selected as shown in the figure 1 "

This is too little about the technique, research methodology and data used.

6) The lack of clearly defined aims of the manuscript makes it impossible to evaluate the conclusions. There are also no references to other studies.

7) The content raises serious objections - is it adequate to the journal's profile? There are other magazines that are more concerned with health protection.

8) I believe that this manuscript has more disadvantages than advantages (one of the advantages is the use of game theory) - I do not recommend it for further work.

Author Response

1) In the introduction, the main goal and specific goals, research problems and research hypotheses were not clearly defined.

Response: Thanks for the suggestion. We’ve revised the paper.

Research objectives: The research objectives of this paper are to reveal the conflicts between care institutions and the families of the disabled elderly using a simplified game model and to analyses the moderating role played by the policy support of government departments. To accomplish this, we have taken three steps: first, we have conducted a survey on the current pilot cities of long-term care in China to enrich our research background and collect research materials for our numerical analysis; second, we have constructed an evolutionary game model to depict the conflict between the disabled elderly and the care mechanism in China and its evolutionary pattern; third, we investigated the internal factors of the long-term care industry and used numerical analysis to investigate the impact of internal factors (cost of care, price of care) and policy support (government subsidies) on the combination of game strategies.

In this study, we present three research hypotheses based on Chinese scenarios and common-sense economics to define the selection strategies of care facilities and families of the crippled elderly (See page 6).

2) The bibliography is quite modest and does not include health care in other countries.

Response: Thanks for the suggestion. Under the Long-Term Care Insurance heading in the literature review, we compare the variations between the nations that are typically represented by long-term care mechanisms, comparing the coverage groups, financial burden, cost control, and staff training of the three different systems.

3) What do A1, A2, A3 mean? In the chapter MODEL AND METHODS? Why did the authors fail to describe and develop these assumptions?

Response: Thanks for the suggestion. The text has been supplemented with descriptions of the three possibilities in greater detail.

“Before constructing an evolutionary game model in order to analyze the game and strategy choices of nursing institutions and families with disabilities, we first make the following assumptions based on China's realistic background and economic common sense.

A1: Assuming that care institution provides two kinds of care services: simple care services and professional care services.

Nursing institutions, as market participants, meet the general characteristics of enterprises. In China's long-term care market, a variety of nursing services can be simply divided into simple care services and professional care services. When providing simple nursing services, nursing institutions can obtain lower business revenue with lower investment costs. Nursing institutions can obtain higher business revenue when providing professional services, but it also means hiring more nursing staff and purchasing more nursing equipment. From the perspective of economics, nursing institutions will choose the most favorable strategy based on cost-benefit analysis.

A2: Assuming that families with disabilities can choose in-home services or in- institution services, both of which supplied by care institution.

The choice of nursing services depends not only on the degree of disability of the disabled elderly, but also on the time cost of care, income status and satisfaction with nursing services of the families with disabilities. Choosing in-home services can also reduce the family's long-term care expenditure, for which family members have to bear a higher cost of care time - which may also become an opportunity cost that restricts the family income level. Choosing in- institution services will reduce the time cost of care for family members; however, high-quality care services can also be obtained by paying higher long-term care expenditure. Families need to consider the above factors comprehensively to choose the most suitable strategy for themselves.

A3: Although the government does not directly participate in the game between the families and the care institutions, it can provide subsidies for the care institutions by taxing all the families.

The operation of long-term care institutions requires a certain amount of nursing staff and nursing equipment in the early stage, which means that there is a certain threshold for the operation of such institutions. In addition, only when the business revenue from nursing services (or the number of disabled elderly people receiving care) reaches a certain level, can nursing institutions balance their income and expenditure and gain profits. Without the policy support of government departments, the operation of nursing institutions may be in trouble. Therefore, in the model analysis of this paper, we assume that the government departments can set different subsidy standards according to the operation mode of nursing institutions.”

4) No review of the analyzed statistical data

Response: Thanks for the suggestion. This paper is a theoretical analysis, and the data source is a national survey on long-term care insurance conducted by Xi'an Jiaotong University. The paper also draws from relevant literature, such as:

[1] Jing Tao, Xing Huixia, Wan Lihong, Qi Iridium. The impact of expanding long-term care insurance pilot on the sustainability of urban workers' medical insurance fund in China[J]. Insurance Research,2020(11):47-62. DOI:10.13497/j.cnki.is.2020.11.004.

[2] Cheng QD. Unbalanced game and response in home care services under long-term care insurance system--an analysis based on care stations in Shanghai[J]. Journal of Xinyang Agriculture and Forestry College,2020,30(01):46-52. DOI:10.16593/j.cnki.41-1433/s.2020.01.010.

[3] Zhang Xiaohan. Exploring the introduction of disability identification rating agencies in long-term care insurance based on game theory perspective[J]. Shanghai Insurance, 2016 (9): 50-52.

[4] Wang Pu-Chi, Lei Yu-Ruo, Lu Pu-Sheng. The dilemma and the way out of China's combined medical and nursing care model[J]. Journal of the National School of Administration, 2018 (2): 40-51.

[5] Shen Yuling, Zhang Junru, Wu Haibo. Analysis of game behavior of stakeholders in long-term care insurance based on incentive compatibility theory[J]. Shanghai Insurance, 2022.

5) It is difficult to say how the research was done to make the data reliable? Are the data samples representative? "Data on elderly people with disabilities and long-term care insurance are primarily drawn from the Chinese Elderly Care Services survey (CECS), which was conducted by School of Public Administration of Xi'an Jiaotong University in 2021. The sample was selected as shown in the figure 1". This is too little about the technique, research methodology and data used.

Response: Thanks for the suggestion. Given the limited nature of the survey data, the analysis results may be influenced. In the ensuing numerical analysis, we did a comprehensive sensitivity analysis, making the study's conclusions more realistic and educational. We examine the evolutionary trend of the game between disabled families and care providers by adjusting variables such as the cost of care services, the price of care services, the cost of family care time, and the government's subsidy tax rate, in order to investigate the conditions for the emergence of win-win strategies.

6) The lack of clearly defined aims of the manuscript makes it impossible to evaluate the conclusions. There are also no references to other studies.

Response: Thanks for the suggestion. We’ve revised the paper.

7) The content raises serious objections - is it adequate to the journal's profile? There are other magazines that are more concerned with health protection.

Response: Thanks for the suggestion. Long-term care is an essential policy component influencing healthy human capital, and healthy human capital accumulation leads to sustained economic development. As a result, our research is consistent with the tone of the journal SUSTAINABILITY. In fact, in reviewing the current literature, we discovered that SUSTAINABILITY had published quite a few articles on long-term care research. As a reference, some of this literature is presented below.

  1. Bogataj, David, Marija Bogataj, and Samo Drobne. 2022. "Long-Term Care Sustainable Networks in ADRION Region" Sustainability14, no. 18: 11154.

https://doi.org/10.3390/su141811154

  1. Saloniki, Eirini-Christina, Agnes Turnpenny, Grace Collins, Catherine Marchand, Ann-Marie Towers, and Shereen Hussein. 2022. "Abuse and Wellbeing of Long-Term Care Workers in the COVID-19 Era: Evidence from the UK" Sustainability 14, no. 15: 9620.

https://doi.org/10.3390/su14159620

  1. Ismail, Mohamed, and Shereen Hussein. 2021. "An Evidence Review of Ageing, Long-Term Care Provision and Funding Mechanisms in Turkey: Using Existing Evidence to Estimate Long-Term Care Cost" Sustainability 13, no. 11: 6306. https://doi.org/10.3390/su13116306

4.Xu X, Chen L. 2019. Projection of long-term care costs in China, 2020–2050: based on the Bayesian quantile regression method[J]. Sustainability 11(13): 3530.

8) I believe that this manuscript has more disadvantages than advantages (one of the advantages is the use of game theory)

Response: Thanks for the suggestion. We appreciate the reviewers' insightful and constructive remarks, which will assist us in revising and enhancing our research findings. We fully considered these revisions in the revised manuscript and made additions and changes to the study background, model construction, and results discussion.

Reviewer 3 Report

The paper approaches in an interesting manner the problem of long-term care for elderly people which becomes more and more an important issue not only for China, but also for many other countries. The authors offer a thorough analysis on this matter, but there are few problems to adjust, as follows:

1. Conclusions can be improved, while they do not point clearly the opinion of the authors on the subject, leaving the readers to draw their own opinions and not offering some suggestions on what should be done either by the care institutions, people or by state.

2. After sustaining in introduction that the authors seek to “find win-win strategies to ensure the effective implementation of long-term care insurance policies”, readers really expect to find such conclusions, but such conclusions do not appear.

3. Authors should review the English writing, while there are several phrases that make no sense (e.g. in page 2 – “Snorre Kverndokk (2021) use Stackelberg game verifies the implications of a fee for the strategic decisions of the hospitals and the long-term care institutions”).

Author Response

The paper approaches in an interesting manner the problem of long-term care for elderly people which becomes more and more an important issue not only for China, but also for many other countries. The authors offer a thorough analysis on this matter, but there are few problems to adjust, as follows:

  1. Conclusions can be improved, while they do not point clearly the opinion of the authors on the subject, leaving the readers to draw their own opinions and not offering some suggestions on what should be done either by the care institutions, people or by state.

Response: Thanks for the suggestion. To improve the paper's policy, corresponding changes were made to the abstract, body, and conclusion sections.

  1. After sustaining in introduction that the authors seek to “find win-win strategies to ensure the effective implementation of long-term care insurance policies”, readers really expect to find such conclusions, but such conclusions do not appear.

Response: Thanks for the suggestion. It is possible to view (professional care, institutional care) as a win-win strategy - a combination in which the health status of the disabled elderly is better maintained, while the long-term care facility earns more money and is in a more developed state. — At this moment, the level of social welfare is much higher than in other combinations.

  1. Authors should review the English writing, while there are several phrases that make no sense (e.g. in page 2 – “Snorre Kverndokk (2021) use Stackelberg game verifies the implications of a fee for the strategic decisions of the hospitals and the long-term care institutions”).

Response: Thanks for the suggestion. We’ve revised the sentences in the paper. “According to Snorre Kverndokk (2021), some countries have implemented a fee to prevent bed blockage in hospitals. This paper introduces the Stackelberg game model, which positions hospitals as leaders and care providers as followers. It then examines the impact of such a fee on the strategic decisions of hospitals and long-term care providers, and theorises that the average level of hospital services will be close to its lowest level prior to the reform.”

Your suggestions helped us revise and improve our work, and they served as a valuable guide for our study. 
